# Exercise Prescription in Renal Transplant Recipients: From Sports Medicine Toward Multidisciplinary Aspects: A Pilot Study

**DOI:** 10.3390/jfmk5010010

**Published:** 2020-01-30

**Authors:** Goffredo Orlandi, Francesco Sofi, Luciano Moscarelli, Lino Cirami, Sabrina Mancini, Laura Stefani

**Affiliations:** 1Sports Medicine and Exercise Unit, Department of Experimental and Clinical Medicine, University of Florence, 50141 Florence, Italy; goffredo.orlandi@icloud.com (G.O.); sabrimanci.72.sm@gmail.com (S.M.); 2Unit of Clinical Nutrition, Department of Experimental and Clinical Medicine, University of Florence, 50141 Florence, Italy; francesco.sofi@unifi.it; 3Nephrology and Dialysis Unit, Clinical and Experimental Department, University of Florence, 50141 Florence, Italy; moscarellil@aou-careggi.toscana.it (L.M.); ciramil@aou-careggi.toscana.it (L.C.)

**Keywords:** unsupervised physical exercise, renal transplant recipients, body composition

## Abstract

Renal transplantation is the choice treatment for end-stage renal disease. In spite of transplantation, cardiovascular morbidity and mortality remains high, possibly due to a prolonged sedentary lifestyle prior to transplantation. This study aimed to evaluate the impact of unsupervised intervention in a tailored home-based aerobic resistance exercise program, based on the anthropometric and cardiovascular parameters in a group of renal transplant recipients (RTRs) followed for 12 months. Methods: a group of 21 RTRs (mean age: 46.8 ± 12 years) were enrolled in a combined aerobic and step count unsupervised prescription program. Body composition (BMI, waist circumferences, skin-folds); water distribution (TBW: Total body water; ECW: Extra cellular water; and ICW: Intracellular water) and myocardial function were measured every 6 months for 1 year. The MEDI-LITE score was used to estimate adherence to the Mediterranean diet. Results: Significant reductions in waist circumference (Waist Cir: 89.12 ± 12.8 cm T0; 89.1 ± 12.5 cm T6 (95% CI: 6.3, 5.7); 88.6 ± 11.4 cm T12; (95% CI: 6.7, 4.7) *p* < 0.01), weight:71.8 ± 14.8 kg T0; 70.6 ± 14.7 kg T6(95% CI:−8, 6); 70.6 ± 14.7 kg T12(95% CI: 6.6, 7) *p* < 0.05), as well as an improvement of myocardial function, as shown by the significant increase of contractility and change in the GLS % value (−18.3 ± 3.8% at T0 (95% CI:−16.57, 20.0.2)−20.4 ± 3.0% at T6(95% CI:−4, 0.2);−22.9 ± 3.1%T12(95% CI:−3, 4, −1, 6) *p* < 0.02), were observed. Adherence to the Mediterranean diet was in the normal range. Conclusions: Despite unsupervised intervention, combined moderate physical exercise appears to have a positive effect on the main parameters related to cardiovascular risk factors. The long-term efficacy of this program requires further investigation, particularly for evaluating constant adherence to the home-based physical exercise program.

## 1. Introduction

Renal transplantation is the choice treatment for end-stage renal disease [1]. It aims to restore a normal lifestyle and prolong survival. Despite transplantation, patients maintain high cardiovascular morbidity and mortality compared to the general population [2]. Due to the use of steroids and immunosuppressive drugs, suboptimal renal function and weight gain are often present in renal transplantation patients [3]. The nutrition in RTR may increase the frequency of infection and accelerate the atherosclerotic process [4]. Moreover, it has long been observed that a sedentary lifestyle is an important factor influencing the increase in cardiovascular mortality, particularly when associated to arterial hypertension, diabetes, and other cardio-metabolic risk factors, such as dyslipidemia [5].

In the field of sports medicine, exercise prescriptions are currently involved in the treatment of many chronic degenerative diseases, such as cancer, hypertension, and diabetes. Individualized physical exercise programs for patients with chronic metabolic diseases are planned and adapted to different contexts [6]. Recently, interest has been extended to transplant recipients of solid organs, particularly among renal transplant recipients (RTRs), promoting a program of physical exercises of moderate intensity [7,8]. Until now, RTRs have been considered a fragile population, with no specific indications for physical activity. The literature more recently reported the positive effects on body composition and quality of life in the case of supervised combined exercise [9]. However, sufficient data are not available for unsupervised home-based programs, and in this context, no international guidelines are available.

Despite the potential and well-known risk of physical exercise as a possible cause of acute events, especially considering possible basic cardiotoxicity due to the use of immunosuppressive drugs, aerobic resistance and moderate individualized counter exercise, established as reported in the ACSM guidelines [10], are now an important and effective therapeutic choice in post-renal transplantation management. The principal aim of this study was to verify the eventual modifications of the anthropometric parameters in RTRs undergoing a program of exercise indications like aerobics and resistance exercise without supervision, for one year. During the follow-up, some other aspects, such as adherence to a correct Mediterranean diet and morphological and functional myocardial parameter, were explored, despite the limited length of the study.

## 2. Materials and Methods

### 2.1. Therapeutic Plan of Intervention: Moderate Individualized Combined Physical Activity and Nutritional Record

The planning of therapeutic treatment with the prescription of physical exercise for RTRs demonstrates that modification of limited physical activity and a sedentary lifestyle can play a decisive role in reducing the mortality and morbidity of cardiovascular events [11,12]. The physical exercise protocol was approved by the local ethics committee as an extension of a previous registered clinical trial for supervised exercise in RTR (ISRCTN66295470) in which a control group doing unsupervised exercise was enrolled. All the subjects investigated gave their consent to participate in the investigation.

Considering this is a pilot study, the RTRs were consecutively enrolled at the Sports Medicine Center of the University of Florence, on the basis of their motivation to adhere to the protocol. They were previously selected at the nephrology unit if they had a kidney transplantation more than 12 months before the start of the study (average time to transplant up to 6 years) and being in a stable clinical condition. Exclusion criteria included anemia, uncontrolled hypertension, symptoms of heart failure, and any other severe restrictions to daily physical activity.

The exercise intervention program included several tests to establish the amount of aerobic and resistance exercise to be prescribed and carried out at home.

Constant adherence to aerobic exercise was aided by a dedicated smartphone application for checking the daily step count and the amount of physical activity.

Cardiovascular evaluation included the ergometric test (EMT) conducted at the maximal tolerated effort to establish the intensity of exercise to be prescribed. A standard 2D echocardiographic evaluation of the systo-diastolic parameters with the addition of the measurement of the deformation parameters as GLS (global longitudinal strain) was also performed. In parallel, nutritional habits were evaluated at every check-up (T0, T6, T12) by the MEDI-LITE score. Body composition was also assessed by muscular strength and flexibility tests and studied by bioimpedance analysis (BIA; BIVA 101-Akern RJL) and skin-fold measurement. All these parameters were essential for planning exercise prescriptions and for follow-up evaluation of subjects, as shown in the flow chart below (Figure 1).

### 2.2. Flow-chart of the Exercise Program Evaluation and Echocardiographic Evaluation.

Starting from previous experience with the home-based exercise program applied with other chronic metabolic diseases [13], an individualized home-based physical activity program was created in the range of moderate intensity (equal to 60−70% compared to maximum effort). Following the ACSM (American College of Sports Medicine) guidelines, the exercise program was composed of combined aerobic and counter-resistance exercises.

A short initial phase of supervised training was given to prevent incorrect exercise.

The program consisted, first and foremost, in assessments and specific clinical sports medicine tests to plan individual aerobic and endurance exercises. Evaluation included the ergometric test (EMT) performed by an Ergoline 200 GmbH-Esaote with an incremental effort of 25 W every two minutes. The hemodynamic blood pressure and electrocardiographic response to exercise was recorded, together with the heart rate (HR), power (W), and rate of perceived effort (CR-10), according to the American College of Sports Medicine guidelines [14].

EMT was interrupted at the maximum tolerated effort. Considering it consisted of progressive incremental workload steps with an increase of 25 W every 2 min until subjective exhaustion, it was not necessary to reach the maximal heart rate. It was performed to establish the intensity of the exercise, in terms of range of heart rate, in order to further propose brisk walking, low jogging, or cycling as a different kind of exercise for home-based training. The range of HR refers to the exercise tolerance scale, as CR10 represents. The aerobic training prescribed included at least 30 min of brisk walking or cycling or some other exercise at 60−70% of the subject’s maximum level. The fatigue scale assessed exercise tolerance, and a value of CR10 of 5−7 was associated to the heart rate level to identify the range of aerobic exercise. This aspect of exercise tolerance is often independent of cardiac performance and therefore it has been related to training and muscle response to acute exercise.

The method was particularly useful as a practical indication for maintaining the same perceived exertion level during the home-based exercise program. This sort of program has been evaluated and validated previously in other contexts of chronic diseases. It is in agreement with the ACSM guidelines [11] and has been adapted recently to the Italian model [12], where home-based management is promoted.

Upper limb strength was measured by the hand grip test. This consisted of a muscle strength test using a hand gripper: Normal values in the range of 30 kg per arm.

Resistance exercises were established, based on the Hand Grip (HG) test. The tests were performed to assess the strength of the upper limbs, generally considered as the expression of global body strength. Following the guidelines of the ACSM [13], resistance exercises included at least 8 groups of muscles without any additional weight to their own. This is a static exercise, performed at a moderate level for at least three times a week, consisting in a program of movements aimed at achieving flexibility and physical fitness.

Body composition was evaluated by measuring anthropometric parameters, such as BMI, waist circumferences, and skin folds. The skin fold measure, as expression of subcutaneous fat, was reported in mm. It was measured at the triceps and biceps levels.

Furthermore, body composition included the water distribution measurement (total body water, extra cellular water, and intracellular water). This aspect has been considered important, especially for those subjects in whom the compartmentalization of body water is frequent [14,15].

Adherence to the exercise program was checked by periodical evaluation through the questionnaire. Subjects were asked to record their weekly physical activity and to report it at every clinical check-up. In addition, a periodical phone call reminder was scheduled at least during the first two months, to maximize awareness of the importance of the program.

### 2.3. Echocardiographic Evaluation

Considering the potential undesirable effects of physical exercise, especially if carried out in an unsupervised manner, heart evaluation was also performed.

This evaluation was carried out through periodic monitoring (at least 3 times T0, T6, T12) with a 2D echocardiogram, by qualified cardiologists using a My Lab 50 echocardiograph (Esaote, Italy), equipped with a 2.5 MHz transthoracic echocardiographic probe. Echocardiographic measurements of the left ventricle (LV) chamber were obtained, following the ACC/AHA guidelines (American College of Cardiology/American Heart Association) [16] from the parasternal long-axis view. The main parameters of the myocardial function were measured (LV end-systolic, LVESd, and end-diastolic diameters, LVEDd; interventricular septum (IVS); posterior wall thickness (PW); and ejection fraction (EF%)). The EF was calculated according to the formula (LVEDd- LVESd/LVEDd), for which volumes are substituted by diameters. Diastolic function was measured by the pulse wave (PW) from the transmitral valve flow velocity and time. For a better estimate of the myocardial contractility, strain analysis was also performed. Strain is a deformation parameter, and the global longitudinal strain (GLS) has especially been demonstrated to have a peculiar sensitivity compared to standard 2D echo parameters in detecting any eventual myocardial impairment. Strain can also be used to eventually implement and increase the intensity of the exercise during training [17,18]. The study of the myocardial function was completed by the LV global longitudinal strain assessment obtained by the software XStrain™— ESAOTE, Genoa, Italy, included in the echo machine. Starting from the echo images acquired at rest (2,3,4 chamber views) and according to the segmentation criteria set in the EACVI/ASE Consensus document [18], strain curves divided into the six segments of the LV chamber were provided. For this investigation, only the GLS parameter, expressed in %, was considered.

### 2.4. Adherence to the Mediterranean Diet

The MEDI-LITE score [19] was used to verify patients’ adherence to the Mediterranean diet that includes nine classes of food, namely fruit, vegetables, cereals legumes, fish, olive oil, meat and meat products, dairy products, and alcohol. The score, ranging from 0 to 18, indicates how often these various foods are consumed.

In parallel, an inverse score was awarded on the basis of a maximum value of 2 for the lowest consumption frequency, 1 for intermediate consumption, and 0 for the highest recruitment category. Particularly, for alcohol, a score was calculated (2 for the intermediate consumption rate, 1 for the lowest intake category, and 0 for the highest consumption dose). The more the score approached the maximum of 18, the higher the adherence to the Mediterranean diet.

## 3. Statistical Analysis

All data are expressed as mean ± standard deviation (SD). Differences between the data of the group followed at three times point were estimated by an independent sample *T*-test for paired data. A repeated-measures ANOVA was used to test the differences between means. *p*-values <0.05 were considered statistically significant. The difference of the values at 95% of the CI was also calculated. The statistical package PASW 20.0 for Macintosh (SPSS Inc., Chicago, IL, USA) was used.

## 4. Results

### 4.1. Population Studied

Among a large cohort of 35 RTRs a restricted cohort of 21 RTRs (mean age: 46.8 +/− 12 years, 14 males and 7 females, similar age range) all adhering and motivated to follow the moderate exercise program without supervision, established individually, were investigated. They were consecutively enrolled and therefore the age range was not considered a specific exclusion criterion. The mean age range from transplantation was 18.4 ± 8 years. The inclusion criteria included having a stable clinical condition and no episodes of rejection in the past. In addition, high-quality echocardiographic images for post-processing analysis of deformation parameters were also required.

All subjects had pharmacological treatment generally consisting of antihypertensives, such as doxazosin, ACE inhibitors, or ARBs. They were controlled well by therapy. An immunosuppressive treatment included cyclosporine or tacrolimus, in combination with mycophenolate or everolimus, and steroids (methylprednisolone). After a first clinical evaluation of a stable clinical condition and positive response to the EMT (flow chart), all RTR subjects concluded the study and no injuries or adverse effects were observed.

The subjects were periodically evaluated for body strength, waist circumference, skinfold measurements, and nutritional aspects, every 6 months for at least 1 year. Constant adherence to the exercise program was verified by the use of dedicated questionnaires, given to patients at each clinical check-up, in which the time dedicated to the activity prescribed was reported. All subjects began with the same exercise program. Participation in the program was confirmed through the use of a specific application for mobile phones, which allowed objectification of the intensity and duration of the weekly activity. All patients were invited to choose their preferred smartphone application, making it easy to check the number of steps per day and the daily amount of physical activity. Follow-up was completed by a nutritional assessment to verify correct adherence to the Mediterranean diet. All subjects were found to adhere to the exercise program prescribed.

### 4.2. Primary Outcomes: Anthropometric Parameters

At baseline, none of the subjects adhering to the prescribed program had acute cardiovascular events and they all concluded the investigation. Body composition expressed by waist circumference and subcutaneous folds showed a significant improvement after 6 months (Table 1)

Despite the fact that skin fold measurements were detected at several points, the values reported as mean SD values refer to the biceps/triceps only.

As shown in Table 1, a significant reduction in weight, waist circumference, and triceps skin fold was observed; on the contrary, BMI, measured by the formula kg/m^2^, did not show significant differences during the protocol.

No substantial variations were observed in the water balance and distribution obtained by the BIA analysis.

The data of the strength test measured by the hand grip test significantly improved at the end of the protocol for the left arm while a trend towards an improvement, in spite of not being significant, was found in the hand grip for the right arm.

### 4.3. Secondary Outcomes

All the standard echocardiographic parameters of the systolic and diastolic function remained normal without any significant changes (Table 2).

The evaluation of cardiovascular performance measured by the deformation parameter, strain analysis, and, particularly, GLS % calculation showed a significant improvement during the follow-up: −18.3 ± 3.8% T0; −20.4 ± 3.0% T6 (95%CI: −4, 0.2), −22.9 ± 3.1% T12 (95% CI: −3, 4, −1, 6) with *p* < 0.02, especially when compared to the left ventricle EF, which did not change significantly (Figure 2).

The analysis of the data by gender showed values within the normal range in the small female group, despite a larger dispersion of the data as expressed by SD. Especially, for the water distribution, the values were not significantly different if compared to the male group (TBW: 41.7 ± 10.6 at T0; 42.6 ± 9.8 at T6 (95% CI: −3.9, 5.7;) 41.5 ± 8.9 at T12 (95% CI: −5.5, 3.3); ECW: 24.3 ± 6.6 at T0; 22.6 ± 8.8 at T6 (95% CI: −5, 2.2); 21.7 ± 7.8 at T12 (95% CI: −4.8, 3) with P:NS). The data obtained from the HG test were, on the contrary, different, especially in the dominant arm and on comparison with the male group (HG dx: 29.2 ± 9.5 at T0; 28.3 ± 9.5 at T6 (95% CI: −5.3, 3.5); 29.2 ± 8.3 at T12 (95% CI: −3.3, 5.1).

Regarding the echocardiographic parameters, with the exclusion of the standard 2D echo data that were normal for age and gender without any modifications during the follow-up, on the contrary, the GLS strain analysis showed a significant improvement in males, older at the time of transplant, if compared to females. The cardiac function by GLS, estimated at the beginning of the protocol, was −17.1 ± 3.7 at T0, −19.2 ± 2.0 at T6 (95% CI: −3.3, −0.7), *p* < 0.05) and remained normal at T 12 −20.5 ± 3.5( 95% CI:−2.1, 1) while in females, it was −19.5 ± 3.9 at T0; −21.8 ± 2.4 (95% CI:−4.1, −0.5) at T6; and −20.7 ± 3.5 at T12(95% CI: −3.4, 1.2) without any significant difference.

Regarding diet evaluation, although no significant differences were found for the MEDI-LITE score, an interesting trend towards major adherence by all subjects at 3 months was observed. No significant variations were found after that period (Table 3).

During follow-up, renal function was found to have remained normal (creatinine (mg/dL) 1.5 ± 0.5 at T0; 1.5 ± 0.7 at T12; Urea (mg/dL) 66 ± 0.26 at T0 up to 71 ± 0.43 at T 12). No episodes of rejection, injuries, or bone fractures were observed. From the data of the questionnaire, where the amount of physical activity was declared, adherence to the training program was excellent: Exercise for at least 130 min per week was declared.

## 5. Discussion

The present study, despite being a pilot study, demonstrates how an unsupervised exercise program can have a positive impact on CV risk factors. The data support the role of exercise intervention in RTR that is a category at high CV risk. It is well known that exercise training is important in kidney transplant patients [20]; however, most of them are not physically active or participate in exercise, despite physicians’ recommendations [21,22,23,24]. Few data are available in terms of specific guidelines to plan an exercise prescription model. Some investigations have been carried out on subjects during dialysis [25], demonstrating how exercise produces positive effects in the 6 Minute Walk Test (6MWT). Some other authors investigated the effects of supervised physical activity [6,7] while a specific experience in the unsupervised exercise program, especially in the context of cardiovascular risk prevention, is missing [21]. Especially, in those subjects with potential high frailty, as in RTRs, the program of investigation needs to be well detailed. The data obtained demonstrate how unsupervised exercise can produce positive effects, especially in those parameters strongly related to cardiovascular events. Particularly, the absence of negative effects on cardiac function and the positive effects on skinfold as well as upper limb strength support the indication to promote this program. The small cohort of the present pilot study does not allow us to deduce any absolute conclusion in terms of the prescription of exercise in RTRs, especially with reference to important aspects, such as when to start an exercise program, and the length of time after a kidney transplant. The latter point is required in order to achieve maximum improvement in training capacity. In both males and females, a positive impact was observed, despite some differences, on all the behaviors of the myocardial function investigated by the GLS analysis. The time to transplant could be an aspect of further investigation in the future, as a potential cause of longer damage exposition.

However, this study offers a series of important ideas that give indications to promote an exercise program in RTRs that is individualized and of moderate intensity. The peculiar aspect of the present pilot study is that of having explored the effects of the exercise program without supervision, as yet not evaluated. Above all, it is important to reduce the trend towards excessive medicalization. It may be reasonable to begin training with an initial period of supervision by specialized personnel in the case of patients presenting sarcopenia or other fragile conditions. During the first phase of the program, performing exercises autonomously can predispose patients to injuries that could compromise the continuation of the program. For this reason, a short period of practice in the gym is essential.

The potential reduction of muscle strength and the control of cardiovascular and metabolic risk are the most important factors for proposing this treatment, in which resistance exercises are aimed at reducing any sarcopenia.

Future studies should include periodic retraining of enlisted and constantly adherent subjects in the gym every 5–6 weeks, in order to adjust the workload based on periodic medical re-evaluation and reduce the risk of injury, tears, or cramps, or in any case disabling the continuation of the program. The analysis of myocardial deformation, as an estimate of contractility and cardiac performance, is another particular aspect that can adequately support the indications for exercise prescription. The preliminary results support the role of GLS in trained RTRs. Any changes in this context with respect to decreasing cardiovascular mortality need to be examined in a larger prospective study and for a longer period of observation.

## 6. Conclusions

The literature reports some of the effects of physical exercise in RTR, focusing on arterial blood anemia, glycaemia, or lipedema, and on several aspects of quality of life. A structured physical exercise program appears to improve aerobic capacity and to ameliorate muscle performance, especially in supervised exercise [26]. Few data are, however, available in terms of the impact of unsupervised exercise. Some authors have stressed the importance of the prevalence of walking and physical activity, over sedentary time [27]. Some others have investigated the barriers to practice physical activity [28]. Some specific aspects of long-term pharmacological treatment have not been deeply evaluated in this category of RTR at high CV risk levels, especially if regular physical activity is allowed. Despite a pilot study with preliminary results, the intention of this investigation was to provide a valid alternative to the continuous and often impossible commitment in the gym, which can lead to excessive medicalization of the patients involved in the program. The program should guarantee a personalized and well-defined treatment tool, in the context of cardiovascular risk and work in the therapeutic field, limiting the potential negative effect.

## 7. Limitations of the Study

The study presents some limitations. Firstly, the sample investigated was small, and adherence to the prescribed physical activity was not correctly followed. In addition, for these reasons, the results cannot be generalized to a large population and more data will be necessary to confirm the results obtained. Some other aspects, such as the potential negative impact on the data obtained due to the different gender of the sample considered, or the multiple drugs taken by the RTR, were not sufficiently investigated, despite the fact that some evidence of a gender difference was highlighted. The conclusions of the present study therefore need to be restricted to this experience and it is not possible, at this moment, to widely share the results in other contexts.

## Figures and Tables

**Figure 1 jfmk-05-00010-f001:**
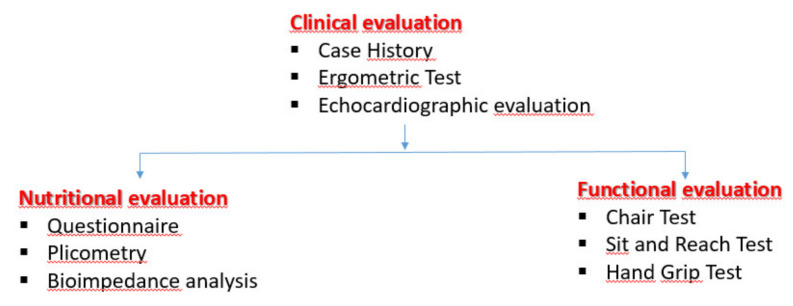
The first step was the clinical evaluation to evaluate the clinical stability and therefore the potential inclusion criteria. Parallelly, nutritional and functional investigations were carried out.

**Figure 2 jfmk-05-00010-f002:**
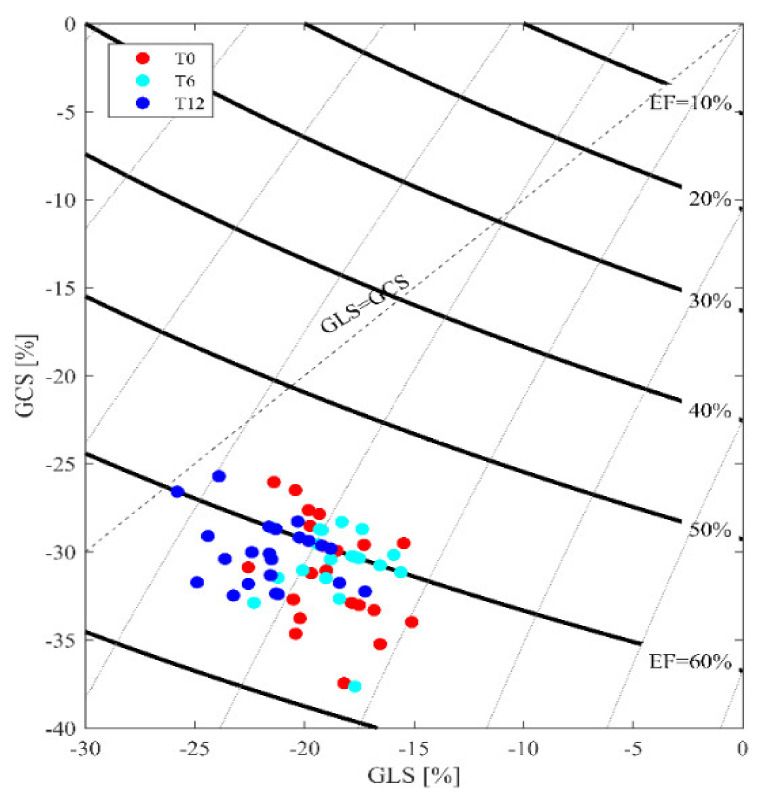
GLS distribution. In the presence of normal and unmodified EF values, increased and more negative GLS values are evident. The cutoff point of the normal range is within the value of −15 up to −17 GLS. The majority of the results are distributed around the −20 GLS value.

**Table 1 jfmk-05-00010-t001:** Anthropometrics and strength parameters of all RTRs during follow-up.

	T0	T6	T12	ΔT0-T6	ΔT6–T12	ANOVA
Weight (kg)	71.8 ± 14.8	70.8 ± 14.3	70.6 ± 14.7 *	1.0 (95% CI: −8.0, 6.0)	−0.2 (95% CI: −6.6, 7.0)	<0.05
Waist Circumference (cm)	89.4 ± 12.5	89.1 ± 12.5	88.6 ± 11.4 *	−0.3 (95% CI: 6.3, 5.7)	−0.5 (95% CI: −6.7, 4.7)	<0.05
Biceps Skinfold (mm)	8.2 ± 3.9	6.9 ± 3.5 *	6.9 ± 3.3	1.3 (95% CI: −05.0, 3.1)	0.0 (95% CI: −0.1, 0.1)	<0.05
Triceps Skinfold (mm)	13.5 ± 3.7	13.6 ± 5.8	11.8 ± 5.1 *	0.1 (95% CI: −2.2, 2.4)	2.0 (95% CI: −0.6, 6.0)	<0.05
BMI kg/m^2^	24.4 ± 2.2	24.2 ± 3.3	24.2 ± 3.1	−0.2 (95% CI: −1.5, 1.5)	0.0 (95% CI: −1.5, 1.5)	0.88
HG right(kg)	32.2 ± 9.6	33.8 ± 10.9	33.9 ±10.8	−1.6 (95% CI: −6.4, 3.2)	0.1 (95% CI: −5.0, 5.2)	0.65
HG left (kg)	33.1 ± 10.4	34.6 ± 11.2	35.7 ± 11.3 *	1.5 (95% CI: −3.6, 6.6)	1.1 (95% CI: −4.1, 6.3)	<0.05
TBW (%)	41.1 ± 9.2	40.6 ± 2.3	40.4 ± 4.2	−0.5 (95% CI: −2.2, 3.2)	−0.2 (95% CI: −1.3, 1.7)	0.58
ICW (%)	20.5 ± 5.0	20.3 ± 4.9	20.3 ± 5.4	−0.2 (95% CI: −2.1, 2.5)	0.0 (95% CI: −2.2, 2.6)	0.55
ECW (%)	23.3 ± 2.6	20 ± 4.5	20.5 ± 5.0	−2.9 (95% CI: −4.6, −1.2)	−0.5 (95% CI: −7.2, −2.8)	0.69

Legend: BMI: Body Mass Index; HG: Hand Grip; TBW: Total Body Water; ECW: Extracellular water; ICW: Intra Cellular Water; * *p* < 0.05.

**Table 2 jfmk-05-00010-t002:** Echocardiographic parameters.

	T0	T6	T12	Δ T0–T6	Δ T6–T12	ANOVA
LVDD	50 ± 2.7	49.7 ± 2.9	50 ± 2.6	−0.5 (95% CI: −1.5, 0.5)	0.5 (95% CI: 0.6, 0.4)	0.74
LVSD	27.8 ± 2.7	28.1 ± 3.9	28.2 ± 2.4	−0.3 (95% CI: −1.8, 1.2)	0.1 (95% CI: −1.4, 1.6)	0.57
EF	63.4 ± 3.4	62.2 ± 4.4	64.4 ± 4.5	−1.3 (95% CI: −3.1, 0.5)	−2.2 (95% CI: −4.0, −0.4)	0.53
E/A	1.14 ± 0.3	1.15 ± 0.5	1.12 ± 0.6	0.01 (95% CI: −0.2, 0.2)	−0.03 (95% CI: −0.2, 0.3)	0.57

Legend: LVDD: Left Ventricle Diastolic Diameter; LVSD: Left Ventricle Systolic Diameter; EF: Ejection Fraction; E/A: E wave A wave ratio.

**Table 3 jfmk-05-00010-t003:** Results of the questionnaires on adherence to the Mediterranean diet measured by the MEDI-LITE score.

	MEDI- LITE T0	MEDI- LITE T6
Mean	11,500	11,700
Standard Deviation	19,003	30,569

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
