# Peer review of "Exercise Prescription in Renal Transplant Recipients: From Sports Medicine Toward Multidisciplinary Aspects: A Pilot Study"

_jfmk, 2020, doi:10.3390/jfmk5010010_

Round 1

Reviewer 1 Report

The paper is improved; however, there are still several issues that need to be attended to. See below for my specific comments.

Line 20: what is meant by “mixed” in mixed unsupervised exercise?

Line 55: Please better define what is meant by “aerobic resistance” and “counter exercise”. Also, that statement needs a citation since it is central to the justification of your study.

Line 63: In the heading for 2.1 it says “nutritional intervention” was it truly an intervention or just a record?

Line 85: The flow chart could be improved, perhaps arrows indicating the order that those tasks were performed would improve it clarity.

Line 167: State reference different groups. What were the groups? If I reading this correctly there is one group with 3 times points that were measured.

Line 168: Why a unidirectional ANOVA?

Figure 1. It is very hard to see how the time point are significant here. Could authors include a line of best fit or something to highlight the difference in the timepoints?

Table 1: Circumference of what?

Line 227-231: What is this? The information seems pertinent, but it doesn’t seem to belong as part of Table 1.  

The discussion was left largely untouched. Based upon the additional references that were highlighted in the previous revision, it was hopefully that the authors would incorporated some of those findings into the present discussion as part of a commentary on how their findings impact the current recommendations and guidelines for exercise in RTR.

Author Response

The paper is improved; however, there are still several issues that need to be attended to. See below for my specific comments.

Line 20: what is meant by “mixed” in mixed unsupervised exercise?

The  text has been  modified

Line 55: Please better define what is meant by “aerobic resistance” and “counter exercise”. Also, that statement needs a citation since it is central to the justification of your study.

 The  intensity of the  exercise  has been  established  following  the ACSM guidelines .It has been  reported in the references .

Line 63: In the heading for 2.1 it says “nutritional intervention” was it truly an intervention or just a record?

 Many  thank to  underline this  aspect. The sentence has been  modified .

Line 85: The flow chart could be improved, perhaps arrows indicating the order that those tasks were performed would improve it clarity.

The flow chart has been improved  following  your suggestion.

Line 167: State reference different groups. What were the groups? If I reading this correctly there is one group with 3 times points that were measured.

  Thank  for  underling  this The sentence has been  modified . We hope now  it is  correct .

Line 168: Why a unidirectional ANOVA?

 The  sentence has been  corrected.

Figure 1. It is very hard to see how the time point are significant here. Could authors include a line of best fit or something to highlight the difference in the timepoints?

 We are in agreement  with  the reviewer . The legend  includes  now  a specific  clarification of  the data.  We hope the figure  is  now  more comprehensible .

Table 1: Circumference of what?

Waist circumference

Line 227-231: What is this? The information seems pertinent, but it doesn’t seem to belong as part of Table 1.  

 Yes  this  is  the table 2

The discussion was left largely untouched. Based upon the additional references that were highlighted in the previous revision, it was hopefully that the authors would incorporated some of those findings into the present discussion as part of a commentary on how their findings impact the current recommendations and guidelines for exercise in RTR.

 The text has been   modified .

Reviewer 2 Report

Thank you for your revised version. 

Despite significant improvements according to reviewer suggestions, I continue to struggle with the setup and presentation of the aim, methods and outcomes of the paper.

The aim is described as a feasibility study.

From the methods section it does not become clear to me what out of the many described tests comprises the intervention, what is an outcome parameter tested in this study and what is the hypothesis. How was feasibility tested and was the desired feasibility reached?

E.g. the Mediterranean diet: what was the intervention on diet or why is this an outcome parameter for feasibility of unsupervised exercise prescription? 

Complete cardiovascular evaluation: is this an in- or exclusion parameter? is this an intervention, a baseline or an outcome parameter? 

In the results section patients were 'invited to choose their preferred 'smartphone application' to check for daily steps and amount of physical activity'. Was this part of the intervention or the outcome? Why was it not described in the methods section?

First sentence results section: is 35 RTR the full cohort of patients of which 21 RTRs are selected? Which cohort is this? Transplanted between when and when in which center? 

Author Response

This piece outlines a research study investigating unsupervised home-based exercise and dietary guidelines for renal transplant recipients. The paper is improved in its revised format as a more standard research piece. The paper remains more difficult to read that would be hoped but written English is improved. The data and findings would be of interest to those working in nephrology or with RTR patients more generally.

Many thank for having appreciated  our paper . We are in agreement  with the reviewer  for the potential use  of the suggestion.

Abstract:

This is a fairly nice summary of the research and is improved by the inclusion of some key findings.

OK thank  for your comment

Introduction:

The introduction provides some context and a basic rationale for the need for the research although this could and should be strengthened. This section could also make better reference to some of the available literature in this field.

 The introduction reports now some other specific reference.

Some statements within this section e.g. RTR’s considered to be a “fragile population” until now are unsubstantiated.

 Thank  you  for  this  suggestion. We have slightly modified  the  text as “potentially” fragile population .

Materials and Methods:

This section is improved and offers a clearer explanation and structure to what was done in this study, in places it is still rather vague around methods/procedures used (Line 104 for example). The flow chart could be improved to clarify in a simple manner the key aspects of the study design and methods. In its current form it is not really adding much to this section.

 The method session  and  the flow chart  for the study design have been detailed.

Further sub-headings would be useful here to aid the clarity of procedures used.

 We have not understand  the  meaning of the question. We  respectfully ask where eventually  the subheadings can be insert .

Justification for methods of choice throughout this section would strengthen the paper and rationale. Why those skinfold sites? How did participants record their weekly physical activity? What are criteria for ‘adherence’?

 The skinfold are one of  the main antropometrics paramenter largely used  to  follow  the impact of the exercise . The exercise follow-up  was possible  by questionnaire report.

Consider statistical analysis as an element of the methods rather than stand alone. The described stats analysis doesn’t represent the methodology outlined above as there has been no mention of a control group yet between groups analysis is outlined. A repeated-measures ANOVA would seem to logical choice here. This has not been re-written as per comments document.

 The text has been modified .

Results:

The first section of the results (population studied) would be better placed as a study design/participants section within the methodology. See previous comment on use of ‘subjects’.

 We apologize , however  we  do  not understand  if  the reviewer prefer  to  substitute  the term of subjects  with partecipants.

Comment relating to skinfolds does not match the sites specified in the methods as you do not report having measured biceps within procedures. Why only report one site when you measured more?

 We   are in agreement  with the reviewer The text  is wrong. It   has been  modified in terms of clarify  as the  measure  of the skin fold  are related  to  the biceps  and triceps  value . 

What are the actual P values? NS or <0.05 doesn’t give us magnitude. This is especially important as you go on to talk about trends towards significance!

 The  significance has  been established  at  this  value ,  despite not very significant .Consider  that  the protocol is planned in an unsupervised program  and therefore  no much  more significance  can be  expected .

Diet. Now refers to T3 which has not been previously mentioned. As per previous comments. This is not significant and doesn’t really add anything. What are you actually trying to infer from this finding?

 The Table was wrong and  now the  time  of investigation is T6.

The GS data is very useful in this context but is glossed over.

 We do  not  understand what  you  mean: The GLS ( global Longitudinal Strain)  and not  GS  have not been  modified absolutely .

Discussion:

Some useful comments related to wider implications of the findings but several statements here are not supported by the current data nor-referenced and seem to be over-reac

This piece outlines a research study investigating unsupervised home-based exercise and dietary guidelines for renal transplant recipients. The paper is improved in its revised format as a more standard research piece. The paper remains more difficult to read that would be hoped but written English is improved. The data and findings would be of interest to those working in nephrology or with RTR patients more generally.

Reviewer 3 Report

This piece outlines a research study investigating unsupervised home-based exercise and dietary guidelines for renal transplant recipients. The paper is improved in its revised format as a more standard research piece. The paper remains more difficult to read that would be hoped but written English is improved. The data and findings would be of interest to those working in nephrology or with RTR patients more generally.

Abstract:

This is a fairly nice summary of the research and is improved by the inclusion of some key findings.

Introduction:

The introduction provides some context and a basic rationale for the need for the research although this could and should be strengthened. This section could also make better reference to some of the available literature in this field.

Some statements within this section e.g. RTR’s considered to be a “fragile population” until now are unsubstantiated.

Materials and Methods:

This section is improved and offers a clearer explanation and structure to what was done in this study, in places it is still rather vague around methods/procedures used (Line 104 for example). The flow chart could be improved to clarify in a simple manner the key aspects of the study design and methods. In its current form it is not really adding much to this section.

Further sub-headings would be useful here to aid the clarity of procedures used.

Justification for methods of choice throughout this section would strengthen the paper and rationale. Why those skinfold sites? How did participants record their weekly physical activity? What are criteria for ‘adherence’?

Consider statistical analysis as an element of the methods rather than stand alone. The described stats analysis doesn’t represent the methodology outlined above as there has been no mention of a control group yet between groups analysis is outlined. A repeated-measures ANOVA would seem to logical choice here. This has not been re-written as per comments document.

Results:

The first section of the results (population studied) would be better placed as a study design/participants section within the methodology. See previous comment on use of ‘subjects’.

Comment relating to skinfolds does not match the sites specified in the methods as you do not report having measured biceps within procedures. Why only report one site when you measured more?

What are the actual P values? NS or <0.05 doesn’t give us magnitude. This is especially important as you go on to talk about trends towards significance!

Diet. Now refers to T3 which has not been previously mentioned. As per previous comments. This is not significant and doesn’t really add anything. What are you actually trying to infer from this finding?

The GS data is very useful in this context but is glossed over.

Discussion:

Some useful comments related to wider implications of the findings but several statements here are not supported by the current data nor-referenced and seem to be over-reac

This piece outlines a research study investigating unsupervised home-based exercise and dietary guidelines for renal transplant recipients. The paper is improved in its revised format as a more standard research piece. The paper remains more difficult to read that would be hoped but written English is improved. The data and findings would be of interest to those working in nephrology or with RTR patients more generally.

Abstract:

This is a fairly nice summary of the research and is improved by the inclusion of some key findings.

Introduction:

The introduction provides some context and a basic rationale for the need for the research although this could and should be strengthened. This section could also make better reference to some of the available literature in this field.

Some statements within this section e.g. RTR’s considered to be a “fragile population” until now are unsubstantiated.

Materials and Methods:

This section is improved and offers a clearer explanation and structure to what was done in this study, in places it is still rather vague around methods/procedures used (Line 104 for example). The flow chart could be improved to clarify in a simple manner the key aspects of the study design and methods. In its current form it is not really adding much to this section.

Further sub-headings would be useful here to aid the clarity of procedures used.

Justification for methods of choice throughout this section would strengthen the paper and rationale. Why those skinfold sites? How did participants record their weekly physical activity? What are criteria for ‘adherence’?

Consider statistical analysis as an element of the methods rather than stand alone. The described stats analysis doesn’t represent the methodology outlined above as there has been no mention of a control group yet between groups analysis is outlined. A repeated-measures ANOVA would seem to logical choice here. This has not been re-written as per comments document.

Results:

The first section of the results (population studied) would be better placed as a study design/participants section within the methodology. See previous comment on use of ‘subjects’.

Comment relating to skinfolds does not match the sites specified in the methods as you do not report having measured biceps within procedures. Why only report one site when you measured more?

What are the actual P values? NS or <0.05 doesn’t give us magnitude. This is especially important as you go on to talk about trends towards significance!

Diet. Now refers to T3 which has not been previously mentioned. As per previous comments. This is not significant and doesn’t really add anything. What are you actually trying to infer from this finding?

The GS data is very useful in this context but is glossed over.

Discussion:

Some useful comments related to wider implications of the findings but several statements here are not supported by the current data nor-referenced and seem to be over-reaching as recommendations based on current findings. Make it clear where you are referring to your own study and where to prior research.

Limitations of this study are presented briefly in relation to low patient number.

Conclusions:

This section is improved in terms of balance around current study but still needs to more clearly draw a conclusion from your current data. Clarity of your take home message is needed. What did you find out that is new? Where next?

hing as recommendations based on current findings. Make it clear where you are referring to your own study and where to prior research.

Limitations of this study are presented briefly in relation to low patient number.

Conclusions:

This section is improved in terms of balance around current study but still needs to more clearly draw a conclusion from your current data. Clarity of your take home message is needed. What did you find out that is new? Where next?

Author Response

Thank you for your revised version. Despite significant improvements according to reviewer suggestions, I continue to struggle with the setup and presentation of the aim, methods and outcomes of the paper.  

The aim is described as a feasibility study. From the methods section it does not become clear to me what out of the many described tests comprises the intervention, what is an outcome parameter tested in this study and what is the hypothesis. How was feasibility tested and was the desired feasibility reached?

1) Thank  you  to  give us    the opportunity to improve  our message.

The  study  has been  conducted in a group  of RTR , in clinical  stable  condition . The  aim of the study is to clarify if an  exercise prescription has positive effects  in case of   unsupervised exercise  .  There  not sufficient data  in literature about this kind  of program.

E.g. the Mediterranean diet: what was the intervention on diet or why is this an outcome parameter for feasibility of unsupervised exercise prescription?

Regarding the mediterranean  diet  , as normally promoted  in a global life style reconditioning ,  the evaluaition of a correct   adherence  to the Mediterranean diet despite in this  case  for a short time , is usually   considered  in order to avoid the potential vanification  of  the exercise intervention . All the  subjects  were adherent to the mediterranean diet   

Complete cardiovascular evaluation: is this an in- or exclusion parameter? is this an intervention, a baseline or an outcome parameter?

 The CV  evaluation has not be  performed to include  or to  exclude the subject , it has been  performed  just  to plan  and to establish the intensity of  exercise program .

 In the results section patients were 'invited to choose their preferred 'smartphone application' to check for daily steps and amount of physical activity'. Was this part of the intervention or the outcome? Why was it not described in the methods section

A dedicated smatphone application  was used  to  to check for daily steps and amount of physical activity'

This  has been  expressed  in the  text , however  no  specific application was  suggested , the subjects  were free to choose  the  preferred .

 First sentence results section: is 35 RTR the full cohort of patients of which 21 RTRs are selected? Which cohort is this? Transplanted between when and when in which center?

 In  the method session  the  description  of  intervention  and the aim of the study has been described . The program of exercise intervention includes several exams  addressed  to establish   an amount of an aerobic  and resistance exercise to prescribe  and to practice  in a home regimen

4)the cardiovascular evaluation included  the EMT . The sentence has been  modified .

 Regarding the  population of RTR selected , the  subgroup  of 21 subjects were selected on the  basis  of  the  better echo image .

Round 2

Reviewer 1 Report

Authors still use the terms “aerobic resistance” and “counter exercise” what are these? Counter to what? There can be resistance in an aerobic activity, but aerobic resistance exercise requires clarification. Without clarification, readers will not be clear on the exercise being prescribed in the intervention.

Thank you for the arrows in the flowchart, but they do not help demonstrate the order in which the tasks were performed and therefore only further confuse the issue.

Figure 1. Authors added a statement below graph, but can anything be included in the graph itself to indicate what is significant and what is not?

Figure 2. Are the commas in the correct location for the Standard deviation of the MEDI-Lite, i.e. 1,9003 as opposed to 19,003. In addition, it appears that these are very large SDs close to two or three-times the mean. Is this normal for that measure?

Author Response

Dear Reviewer,

Thank you for your revision of our paper and for indications on how to improve the  manuscript. However, I must point out that to receive at this stage (after the 4th revision!), a request for information that frankly could have been made before, is in our opinion unreasonable:  our aim in this paper is to present a clinical message,  not to pride ourselves on being  exhaustive in this context.   Besides, you yourself make the criticism that this is a pilot study and then suggest we provide data that perhaps not even 200 cases would enable us to present.

Regarding some points raised, e.g.:…. Did you test for the affects of Age and Sex? Despite the effort randomize equally, you cannot be sure that these factors did not influence your analysis. Given the age of SD of your cohort women will be pre/post/and peri-menopausal…..I  respectfully  underline that the  sample  investigated  is small, and  the analysis  you suggest  does not come within the principal aim of this  study. The  paper is a pilot  study, as you  point out more than once.  In my opinion, in order to bring the  eventual impact  of gender differences to emerge in the  results it  is fundamental  to have more data. The paper was not planned  to be completely  exhaustive  in this field. It is addressed  to  draw attention to a potentially  new  clinical approach in these subjects.

[1] If you are indeed performing an exploratory study then I would suggest informing the reader from the start in the title. "Exercise Prescription in Renal Transplant Recipients: from Sports Medicine Toward Multidisciplinary Aspects: An Exploratory Study."

 The title has been  modified

[2] When reporting your data, pre and post values should be reported as mean (SD). Change data, whether percent or mean should be reported as (change, 95%, CI, xx, yy).

 The table has been  modified following  the main suggestions.  

[3] Though the journal does not require it, I would suggest adopting a Consort Reporting style by adding the following: (1) The delineation of a Primary Outcome (what singular variable was the most important to your study and Secondary Outcomes (those that would support or add pertinent information to your findings and (2) a specific and direct hypothesis targeting these outcomes. You may think this as pedantic, but an "aim" is an aspiration and a hypothesis tells your readers that you have thought specifically about what you investigated. If your outcomes were not prioritized then your study may itself to spurious statistical findings, though one could create a referenced objection to this given the small number of comparisons.

That said, I think this paper qualifies as a Pilot Study which is defined as: "A pilot study is a preliminary small-scale study that researchers conduct in order to help them decide how best to conduct a large-scale research project."
In essence it is a feasibility study with no structuring of outcomes. You may report outcomes but should not offer more than what you observed and emphasize the study feasibility within you paper.
We  have  modified the text following  your suggestions . We respectfully underline that , as a consequence of   the limits  of  the  study , this paper cannot  be  considered a study to be  completely exhaustive .

[4] Adopt a Consort Diagram to outline your study flow.The image of  the study flow has been modified, however it  is  a simple flow-chart to explain  the  sequence  of  the   tests used  in case of inclusion in the  study . We normally use  the  flow-diagram in case  of selection of many CASES studied, selected  in terms  of single or multiple parameters, as in the case of review  or metanalisys. This  is  a pilot  study  and therefore, with  exclusion of the inclusion criteria to enrol the patients, and linked  to  the  clinical  evaluation , no other  aspects can be  included in an eventual  decisional plan .

[5] Add a Table - Preferably Table 1 - Describing the basic demographics and health parameters obtained during the study. Age, height, weight, BMI, Waist Circumference, etc.

Considering  that age and height do not change  during  the 12 months  of  observation, we  preferred to insert  in the  same  existing  table only    weight . We reasonably  think  that this is the   only parameter  that can  give  more   weight to  the aim of the study . The  other data  are included  in the  text. .

[6] Give careful consideration to factors that may need to be factors that might influence your analysis. Did you test for the affects of Age and Sex? Despite the effort randomize equally, you cannot be sure that these factors did not influence your analysis. Given the age of SD of your cohort women will be pre/post/and peri-menopausal. 

The  study  is  a pilot study  and the small sample  does not allow for this . Thank  you  for underlining  this  aspect . This   has been discussed  in the dedicated  section.
[7] Create footnotes for each Table defining abbreviations. Also add something to the effect of: "Data reported as Mean (SD) and mean or percent change (95% CI) when appropriate. Each table will be slightly different.

The table cointains acronim  explained in a specific legend. The perimenopausal time, considering  the  small sample  of   women, has not  been  considered  a principal aspect. This could  be  very  interesting in a large sample.

[8] Line 182. How did you approach your post-hoc assessment for between differences between T0, T6, T12? Did you adjust for multiple comparisons? 
T0 vs T6
T6vs T12
T0 vs T12

That's three comparisons or a requisite p-value of 0.017 for significance.
You could use a Dunnet-Hsu to compare T6 and T12 to baseline (0.025)
Or, you could look at change from T0 to T6 and T0 to T12 and use the 95% CI to determine significance. 95% CI that crosses zero = not significant; 95% CI does not cross zero = significant.

Regarding   Comment n8,  analysis has been  conducted to verify  the statistical differences, and not  as  multiparametric analysis. In any case the differences have been  evaluated comparing the data during the  follow-up and also among  the different times  Considering  the small sample  investigated  and  the restricted message, no other statistical analysis has been conducted .

Therefore, perform a GLM analysis in SPSS, choose Age and Sex as covariates and report their statistical values.
AS previously specified  the GLS  is  an additional  parameter to confirm  the normal myocardial  function . No other statistical  investigation has been  performed

[9] Please report your effect sizes for your analyses. For an ANOVA, partial eta squared would be the correct reporting and the interpretation of these should be defined for the readers in the Statistics section. Your SPSS outputs should already contain these values. 

As previously reported the  data have been  analysed following your 3 revision

[10] Table 1 makes no sense as you have only provided the overall p-value. Well, that is how it reads. Did you perform a post-hoc assessment and were there treatment differences?

Tab 1 now  reports  the  ANOVA test .

[11] Can you report on the length of time participants were undergoing treatment or the length of time they were on medication? If so, I would suggest considering testing to see if this influenced your findings.

Post transplant  pharmacological treatment normally starts  with  the  surgical  treatment, for all the  time .This  aspect, and  in addition  the missing data  in literature regarding  the potential  damage especially  if exercise  is  practiced, are  cited  in the  Discussion.  

[12] In the first paragraph of our Discussion you should take the time to accept or reject your hypotheses.

 The initial part of the discussion has been modified .

[13] Create a penultimate paragraph the honestly assesses the limitations of your study and who the results can or cannot be generalized to. 

The limits of the study have been detailed .